## RESEARCH ARTICLE

# Flow similarity model predicts allometric relationships among *Acer platanoides* L. branches

**Charles A. Price**[1,2]*, **Gregory A. Dahle**[3], **Jason C. Grabosky**[4]

**1** Ecology and Evolutionary Biology, University of Tennessee, Knoxville, Tennessee, United States of America, **2** National Institute for Modelling Biological Systems, University of Tennessee, Knoxville, Tennessee, United States of America, **3** School of Natural Resources & the Environment, West Virginia University, Morgantown, West Virginia, United States of America, **4** Department of Ecology, Evolution and Natural Resources, Rutgers University, New Brunswick, New Jersey, United States of America

\* cprice9@utk.edu

## Abstract

Using physical models to predict patterns of plant growth has been a long-standing goal for biologists. Most approaches invoke either thermodynamic, biomechanical or hydraulic principles and assume the mechanism of interest applies similarly throughout the plant branching architecture. A recent effort, the flow similarity model, predicts numerous aspects of branching physiology and morphology and argues that the physiological constraints experienced by plants change as a function of branch order and size, with more basal portions satisfying more biomechanical constraints, and more distal portions, hydraulic ones. Distal branches are expected to have a strong influence on allometric relationships within plants due to their numerical abundance. Here we evaluate the predictions of the flow similarity model and a well-known alternative fractal branching model, using data on the dimensions of 3,484 individual stem internodes across four individual *Acer platanoides* trees. Overall, we find strong agreement between model predictions and the allometric exponents describing tree branch allometry. Further the predicted curvature in allometric relationships is found in all 24 cases examined and the frequency distributions of branch lengths and diameters are consistent with model expectations in 6/8 cases. We also find the area ratios are consistent with the model assumption of area-preserving branching. Collectively, our data and analysis provide strong support for the flow similarity model, and identifies several areas in need of subsequent inquiry.

**Data availability statement:** Data are available in an online repository: https://doi.org/10.7282/00000503

**Funding:** The author(s) received no specific funding for this work.

**Competing interests:** The authors have declared that no competing interests exist.

## Introduction

Understanding how the physical constraints experienced by plants shapes their growth within and across species has been a long-standing goal of biology. A number of prominent modelling attempts based on hydraulics and/or biomechanics have been put forth to explain how different aspects of plant physiology and morphology scale with plant size. A few models stand out due to the attention they have received. The pipe model introduced theoretical links between plant hydraulic architecture and biomass partitioning [1]. The elastic similarity model predicts the ratio of a branches length to its deflection will remain constant across branches of differing size and has been used to predict that branch length will scale with radius with an exponent of 2/3 [2]. More recently, West et al. [WBE, 3,4] have argued that "energy minimization" has led to a fractal branching structure which they maintained applied across both plants and animals, and assumes that branching networks follow elastic similarity. All three of these models have received considerable attention with mixed results that are beyond the scope of our paper to evaluate. The following articles can serve as jumping off points for this rich body of work [5–8]. Additional models based on thermodynamics have been published that also warrant consideration [9,10]. A summary view is that no single model has yet been able to capture the central tendency and variance found within and across plant allometric relationships, where allometry refers to different rates of growth in different body parts [11].

More recently Price et al. published their "flow similarity" model as an alternative to these previous offerings [12]. The model is based on the behavior of fluid flow in bifurcating idealized conduits and describes numerous aspects of plant branching geometry, topology, allometry and physiology. The model is based on two parsimonious underlying assumptions. The first is area-preserving branching in which the ratio of the total cross-sectional area of daughter branches (subscript $D$) to that of the parent branch (subscript $P$) on either side of a branch bifurcation equals one, $\sum \pi r_D^2 / \pi r_P^2 = 1$. Area-preserving branching has been known since as far back as DaVinci, and is sometimes referred to as DaVinci's rule. It has been tested in tree branches [12–14] and leaf veins [15], found to be robust, and has been adopted by other prominent models [1,4]. The second assumption is simply the conservation of volume between the fluid flowing from parent to daughter branches which supplies metabolically active tissue in vascular cambium and leaves. From these two assumptions a number of predictions emerge for the scaling relationships between different linear dimensions of plant parts [Table 1, 12].

As biomechanical demands become large in the form of wind, snow or self-loading, the flow similarity model also predicts that scaling relationships for the basal parts of the tree network will shift, trending toward slope values closer to those predicted by the elastic similarity model. The argument is based on the idea that basal parts (trunk, large branches) of the tree branching network serve a different role with different physical demands than terminal branches, with basal parts requiring more biomechanical investment and terminal parts more consistent with expectations based on hydraulics.

**Table 1. The third column contains the expression that relates the variables listed in the first and second columns. The fourth and fifth columns contain the expected exponents under flow (*CFS*) and elastic similarity models, respectively. Note that the exponent for the relationship between length and diameter (α) determines the values of the other exponents. The 6ᵗʰ column the expected curvature when plotting the relationship in the third column.**

| Y-variable | X-variable | Expression | Flow Similarity (*CFS*) | Elastic Similarity (WBE) | Expected Curvature (*FFS*) | Figure |
|---|---|---|---|---|---|---|
| Surface Area | Volume | $SA = V^{(\alpha+1)/(\alpha+2)}$ | 3/4 | 5/8 | Concave | 1A |
| Length | Diameter | $L = D^{\alpha}$ | 2 | 2/3 | Concave | 1B |
| Diameter | Volume | $D = V^{1/(\alpha+2)}$ | 1/4 | 3/8 | Convex | 1C |
| Length | Volume | $L = V^{\alpha/(\alpha+2)}$ | 1/2 | 1/4 | Concave | 1D |
| Diameter | Surface Area | $D = SA^{1/(\alpha+1)}$ | 1/3 | 3/5 | Convex | 1E |
| Length | Surface Area | $L = SA^{\alpha/(\alpha+1)}$ | 2/3 | 2/5 | Concave | 1F |

Going from the largest to smallest branches within a tree [16], or across trees of differing size [17], we might see a shift in allometric exponents. However, when looking across all branches within an individual tree, terminal parts of the network (small branches, twigs) are still expected to have a strong influence on observed global scaling exponents, such as whole tree metabolism, due to their numerical dominance. For example, there is typically only one internode (the section between branching nodes) the size of the bole in a tree, but there could be tens of thousands of terminal branches of similar size which will exert a strong influence on fitted regression slopes involving internode linear dimensions, or measures of entire tree properties (mass, linear dimensions, resource use). Predicting the exact function describing how the scaling exponents might shift going from large to small branches is challenging due to the wide variety of side branching topologies found in plants [18], however, the approximate range of slopes and the form of the expected curvature (concave or convex), is predictable [12]. To distinguish between these two scenarios, we highlight that the *constrained flow similarity* predictions (Table 1) are for the more terminal portions of the network, and the predictions from the *full* model are the non-linear predictions, the concavity/convexity and slope range. We will refer to these different predictions as *constrained flow similarity* (*CFS*) and *full flow similarity* (*FFS*) throughout.

Here we utilize a dataset containing measurements for 3,484 individual internodes across four individual trees of the species *Acer platanoides*, collected as part of an analysis of allometric patterns in tree branches [16]. Dahle and Grabosky examined how the scaling exponents describing the relationship between branch length and diameter changed going from small to large branches and noted that above a size threshold (3m length), the fitted slopes were consistent with the elastic similarity model predictions. Detailed data on the dimensions (length, radius) of tree branches throughout a tree's architecture are not commonly collected, thus this dataset allows us to evaluate a number of the flow similarity model's predictions, specifically:

1) Allometric slopes fit across all branches in a tree will be closer to values predicted by the *CFS* model due to the numerical dominance of small branches (Table 1).

2) Relationships will have the predicted curvature (*FFS*, Table 1).

3) The ratio of the sum of daughter branch to parent branch cross sectional area will have a mean of 1 (*FFS* assumption).

4) The length distribution will be better fit by an exponential model.

5) The diameter distribution will be better fit by a power law model.

Evaluating scaling models is an area worthy of careful consideration [6,7]. When two models make different predictions the best way to determine support is through model competition, with penalties for the number of parameters if

appropriate [19]. We employ such an approach here, evaluating the predictions of both the *CFS/FFS* models and the WBE model in those cases where comparable predictions exist. This approach follows the methods of multiple working hypotheses [20] which has been advocated for studies of biological scaling [21,22].

## Materials and methods

Branches were collected from the exterior portion of four *Acer platanoides* plantation trees growing at Rutgers University Horticultural Farm III (40°27'44" N, 74°25'44" W) in New Brunswick New Jersey, USA. All data was collected during the summers of 2005 and 2006. DBH values for the four trees ranged from 24.4 to 41.8 cm. Branch length was measured using a string to follow the contour. Branch diameters were measured prior and distal to each bifurcation point using calipers. Branches less than 100 mm in length or 1.5 mm in diameter were not measured. See Dahle and Grabosky [16] for additional details pertaining to the field collection procedures, and raw data can be found at: [23]. The surface area and volume for each branch was calculated from diameter and length using standard formula based on the assumption that each branch can be approximated as a cylinder.

Allometric relationships were examined amongst the four, dimensional variables; length (*l*), diameter (*d*), surface area (*SA*), and volume (*V*), in pairwise combinations as described in Table 1. Standardized major axis (SMA) regression slopes, confidence intervals, and intercepts were determined for both intraspecific and interspecific relationships using the software package SMATR [24]. SMA is typically used in allometric relationships where there is measurement error in both variables [11,25]. Data were log transformed prior to fitting regression slopes to meet the assumption of homogeneity of variance. To compare the difference between model predictions and observed results, we calculated the root mean squared error (RMSE) for each model exponent prediction (Table 2).

To test the direction of curvature in each allometric relationship, we fit a second order polynomial function to each bivariate relationship on log-log axes, then recorded the sign of the second derivative of that polynomial which indicates if the curve is concave (sign is negative) or convex (sign is positive).

To examine daughter to parent branch areas and test the assumption of area-preserving branching, we simply used the radius of each to calculate area, and looked at the ratio of the two, $(\pi r_{D1}^2 + \pi r_{D2}^2)/\pi r_P^2$, where the subscripts refer to daughter (*D*) and parent (*P*) branches, respectively. If the ratio equals one the measured areas are equal.

We looked at the distributions of lengths and diameters for each trees data, by fitting both exponential and power functions to each data array. To compare the exponential and power-law model fits, we used maximum likelihood to estimate the model parameters and likelihood. We then used Akaike's information criterion with correction for sample size, to compare models, $AICc = AIC + (2k(k+1))/(n-k-1)$ [19], where *k* is the number of model parameters, which is 1 for the exponential model, and 2 for the power-law model.

## Results

The mean $R^2$ across all tree level regressions was 0.942 (Fig 1, Table 2). The relationship with the highest mean $R^2$ was *SAvV* (0.995). The relationship with the lowest mean $R^2$ was *LvD* (0.836). The observed allometric slopes were closer to the *CFS* model predictions in all cases (Table 2), and the RMSE was lower for the *CFS* model in all cases, both within and across individual trees (Table 2).

The tests for concavity/convexity (Table 1) were consistent with the flow similarity model prediction in all cases.

The distribution of area ratios was centered on a value of 1 across all trees (Fig 2A) and within each individual tree (Figs 2C–F). All of the distributions were leptokurtic (kurtosis>3). The variance in area ratio increases with decreasing branch size (Fig 2B).

Table 3 reports corrected AIC scores (*AICc*) for the exponential and power-law models and their relative likelihood, which is the probability that the model with the lower *AICc* score minimizes information loss [19]. Six out of eight of the length and diameter distributions were consistent with model predictions (Fig 3). All four of the branch length distributions

**Table 2. Observed relationships between branch dimensional variables.** The first and second columns contain they and X variables respectively. Columns 3-13 contain in order; the grouping (all trees or tree number), the sample size (n), correlation coefficient ($R^2$), the slope, lower and upper slope 95% confidence intervals, the intercept, lower and upper intercept 95% confidence intervals, and the RMSE for the *CFS* and WBE models. Note the RMSE values are lower for the *CFS* in all cases.

| Y variable | X variable | Tree | n | $R^2$ | Slope | LowCI | UppCI | CFS | WBE | Interc | LowCI | UppCI | CFS RMSE | WBE RMSE |
|---|---|---|---|---|---|---|---|---|---|---|---|---|---|---|
| Surface Area | Volume | all | 3484 | 0.995 | **0.756** | 0.754 | 0.758 | **0.750** | **0.625** | 0.742 | 0.735 | 0.750 | 0.006 | 0.131 |
| Length | Diameter | all | 3484 | 0.832 | **2.060** | 2.032 | 2.088 | **2.000** | **0.667** | 0.707 | 0.682 | 0.733 | 0.060 | 1.393 |
| Diameter | Volume | all | 3484 | 0.955 | **0.252** | 0.250 | 0.254 | **0.250** | **0.375** | −0.171 | −0.179 | −0.163 | 0.002 | 0.123 |
| Length | Volume | all | 3484 | 0.957 | **0.519** | 0.515 | 0.522 | **0.500** | **0.250** | 0.355 | 0.340 | 0.370 | 0.019 | 0.269 |
| Diameter | Surface Area | all | 3484 | 0.921 | **0.333** | 0.330 | 0.336 | **0.333** | **0.600** | −0.418 | −0.431 | −0.406 | 0.000 | 0.267 |
| Length | Surface Area | all | 3484 | 0.981 | **0.687** | 0.683 | 0.690 | **0.667** | **0.400** | −0.154 | −0.166 | −0.142 | 0.020 | 0.287 |
| Surface Area | Volume | 1 | 1016 | 0.996 | **0.745** | 0.742 | 0.748 | **0.750** | **0.625** | 0.793 | 0.780 | 0.806 | 0.005 | 0.120 |
| Surface Area | Volume | 2 | 1554 | 0.995 | **0.761** | 0.758 | 0.764 | **0.750** | **0.625** | 0.731 | 0.719 | 0.742 | 0.008 | 0.128 |
| Surface Area | Volume | 3 | 502 | 0.995 | **0.755** | 0.751 | 0.760 | **0.750** | **0.625** | 0.728 | 0.709 | 0.747 | 0.008 | 0.129 |
| Surface Area | Volume | 4 | 412 | 0.995 | **0.763** | 0.757 | 0.768 | **0.750** | **0.625** | 0.694 | 0.671 | 0.718 | 0.009 | 0.131 |
| Length | Diameter | 1 | 1016 | 0.870 | **1.899** | 1.857 | 1.941 | **2.000** | **0.667** | 0.870 | 0.830 | 0.910 | 0.101 | 1.232 |
| Length | Diameter | 2 | 1554 | 0.821 | **2.137** | 2.092 | 2.182 | **2.000** | **0.667** | 0.675 | 0.637 | 0.714 | 0.120 | 1.356 |
| Length | Diameter | 3 | 502 | 0.837 | **2.054** | 1.982 | 2.128 | **2.000** | **0.667** | 0.648 | 0.582 | 0.714 | 0.103 | 1.367 |
| Length | Diameter | 4 | 412 | 0.815 | **2.170** | 2.081 | 2.262 | **2.000** | **0.667** | 0.534 | 0.450 | 0.617 | 0.123 | 1.402 |
| Diameter | Volume | 1 | 1016 | 0.968 | **0.261** | 0.258 | 0.264 | **0.250** | **0.375** | −0.215 | −0.228 | −0.202 | 0.011 | 0.114 |
| Diameter | Volume | 2 | 1554 | 0.950 | **0.248** | 0.245 | 0.250 | **0.250** | **0.375** | −0.161 | −0.172 | −0.150 | 0.008 | 0.121 |
| Diameter | Volume | 3 | 502 | 0.956 | **0.252** | 0.248 | 0.257 | **0.250** | **0.375** | −0.156 | −0.175 | −0.136 | 0.007 | 0.122 |
| Diameter | Volume | 4 | 412 | 0.947 | **0.246** | 0.240 | 0.251 | **0.250** | **0.375** | −0.127 | −0.151 | −0.104 | 0.006 | 0.124 |
| Length | Volume | 1 | 1016 | 0.965 | **0.495** | 0.490 | 0.501 | **0.500** | **0.250** | 0.462 | 0.436 | 0.487 | 0.005 | 0.245 |
| Length | Volume | 2 | 1554 | 0.956 | **0.529** | 0.524 | 0.535 | **0.500** | **0.250** | 0.332 | 0.309 | 0.354 | 0.021 | 0.263 |
| Length | Volume | 3 | 502 | 0.959 | **0.518** | 0.509 | 0.527 | **0.500** | **0.250** | 0.328 | 0.289 | 0.367 | 0.020 | 0.264 |
| Length | Volume | 4 | 412 | 0.955 | **0.534** | 0.523 | 0.545 | **0.500** | **0.250** | 0.257 | 0.211 | 0.304 | 0.024 | 0.269 |
| Diameter | Surface Area | 1 | 1016 | 0.943 | **0.350** | 0.345 | 0.356 | **0.333** | **0.600** | −0.493 | −0.514 | −0.472 | 0.017 | 0.250 |
| Diameter | Surface Area | 2 | 1554 | 0.913 | **0.326** | 0.321 | 0.330 | **0.333** | **0.600** | −0.399 | −0.417 | −0.380 | 0.013 | 0.262 |
| Diameter | Surface Area | 3 | 502 | 0.923 | **0.334** | 0.326 | 0.342 | **0.333** | **0.600** | −0.399 | −0.430 | −0.367 | 0.011 | 0.264 |
| Diameter | Surface Area | 4 | 412 | 0.909 | **0.322** | 0.313 | 0.332 | **0.333** | **0.600** | −0.351 | −0.388 | −0.314 | 0.011 | 0.267 |
| Length | Surface Area | 1 | 1016 | 0.984 | **0.665** | 0.660 | 0.670 | **0.667** | **0.400** | −0.066 | −0.087 | −0.045 | 0.002 | 0.265 |
| Length | Surface Area | 2 | 1554 | 0.981 | **0.696** | 0.691 | 0.700 | **0.667** | **0.400** | −0.176 | −0.194 | −0.158 | 0.020 | 0.281 |
| Length | Surface Area | 3 | 502 | 0.982 | **0.686** | 0.677 | 0.694 | **0.667** | **0.400** | −0.171 | −0.202 | −0.140 | 0.020 | 0.282 |
| Length | Surface Area | 4 | 412 | 0.981 | **0.699** | 0.690 | 0.709 | **0.667** | **0.400** | −0.228 | −0.265 | −0.191 | 0.024 | 0.287 |

were consistent with the exponential model. Two of the four branch diameter distributions were consistent with the power law model, while the other two were better fit by the exponential model.

## Discussion

We have shown that the scaling exponents describing the allometry of branch dimensions within *Acer platanoides* trees are far more consistent with the predictions of the *CFS* than WBE, both within and across individuals. In every case we examined the fitted allometric exponents were closer to the predictions of the *CFS* model than WBE with lower RMSE values (Table 2). Further the curvature (concave or convex) in all six relationships across all four trees is consistent with *FFS* predictions, and the predicted frequency distributions of branch lengths and diameters are consistent with flow similarity

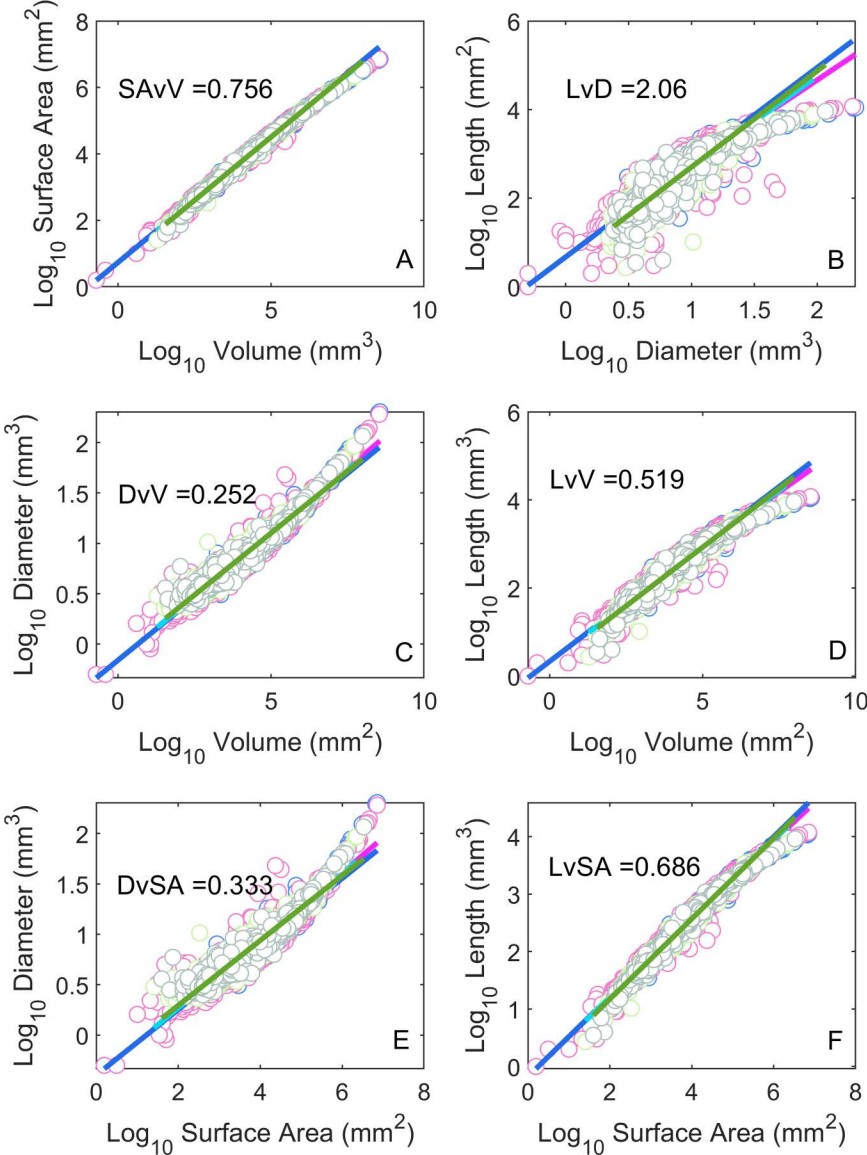

**Fig 1. Allometric relationships between the tree branch dimensions as depicted in Table 1.** The mean slope across all four trees is shown in each panel. Summary regression statistics are given in Table 2. Note the data curvature (concave or convex) in each panel is consistent with the prediction in Table 1.

expectations in six out of eight cases (Table 3). Collectively our analysis of the Dahle and Grabosky data [16] provide strong support for the flow similarity model as a useful abstraction of tree branching architecture.

The *FFS* assumption of area preserving branching is also strongly supported by the data (Fig 2A). This is consistent with what other investigators have found [12–14], and strengthens the case that area-preserving branching is an organizing principle in plant branching networks. A bifurcating fluid flow network that preserves cross sectional area also preserves flow velocity on either side of a branching event. Whether the conservation of flow velocity, or some other physiological trait is the constraint natural selection is acting upon, remains to be determined. As seen in Figs 2A–F, the distribution of area ratios is modal and leptokurtic. While the central tendency is clearly centered on an area ratio of one

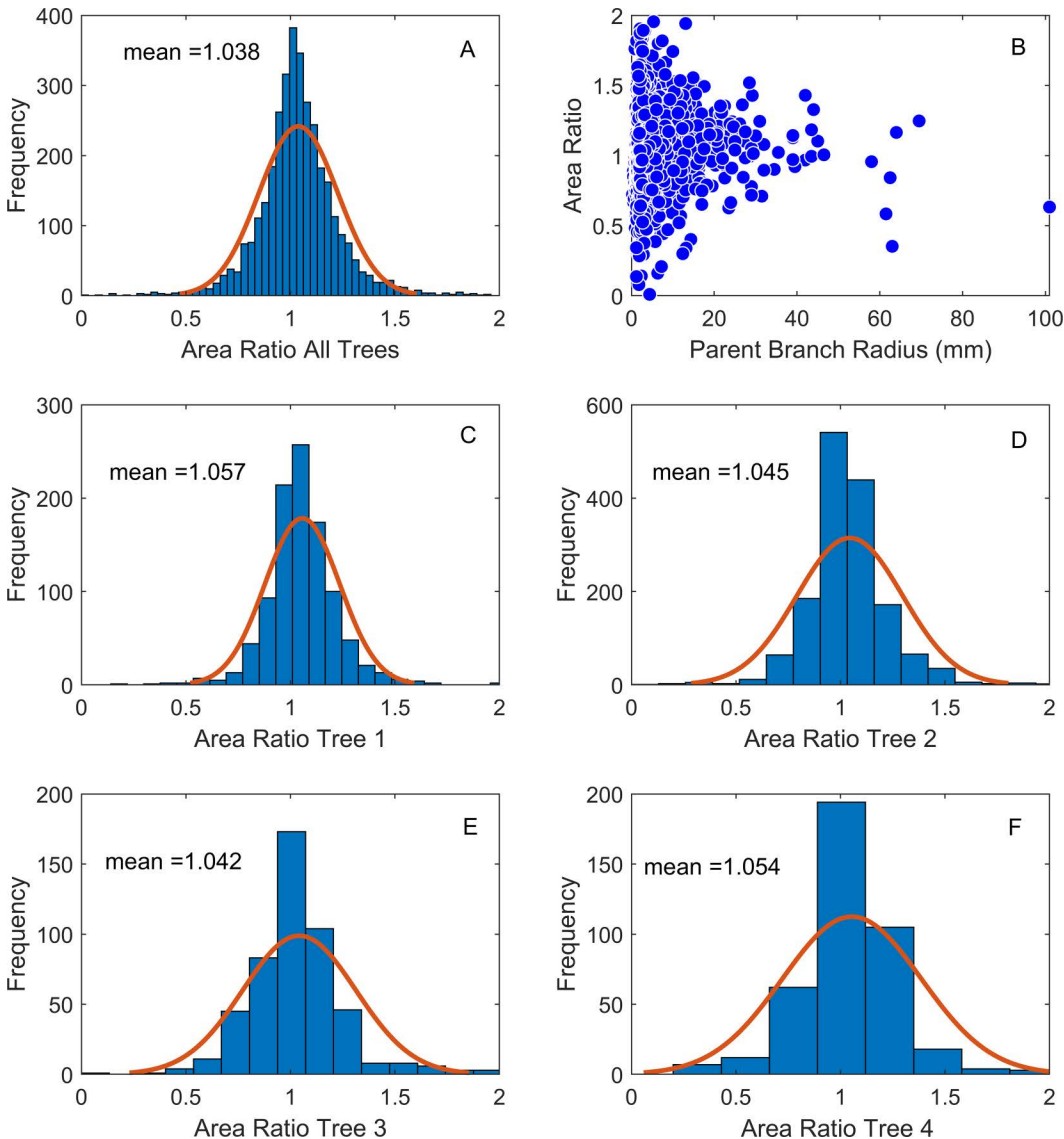

**Fig 2. Frequency distribution of area ratios across all trees (Panel A) and for each individual tree (Panels C-F).** The area ratio plotted as a function of the parent branch radius is in Panel **B.** Note the increasing variance in area ratios with decreasing branch radius (see Discussion).

(mean = 1.038), there exist many values above and below the mean. Moreover, the variability increases with decreasing branch radius (Fig 2B). It is easy to imagine higher variance in measurement among small branches due simply to their small size. However, whether the observed variation is due to measurement accuracy or natural variation is unknown and a target for future investigations.

On the continuum of global tree size, the four trees we examined are on the medium-small end of the spectrum, with DBH's ranging from 24.4–41.8 cm. This may explain why the scaling exponents in Table 2 are so close to the *CFS* model predictions. As mentioned in the introduction, as trees become larger, the *FFS* model predicts that basal parts of the tree network will bear increasing biomechanical loads with scaling exponents shifting *toward* values more consistent with elastic similarity. Indeed, in the original analysis of the data contained in this paper, exponents fit to a subset of the data,

**Table 3. Each tree was tested to determine if the distribution branch segment lengths (rows 2-5) and diameters (rows 6-9) were better fit by an exponential model or a power-law model (highlighted in bold). In all cases, length distributions were better fit by an exponential model. In two of four cases the diameter distributions better fit by a power law, and the other two an exponential. The relative likelihood is given in column 3. Columns 4 and 5 represent the size corrected Akaike's information criterion score for the power law and exponential models respectively, followed by the sample size in column 6.**

| Tree # | Measure | Relative Likelihood | *AICc* Power Law | *AICc* Exponential | SampleSize |
|--------|---------|---------------------|------------------|--------------------|------------|
| 1 | Lengths | 1.01E-182 | 16699.060 | **15860.929** | 1016 |
| 2 | Lengths | 0 | 25533.535 | **23461.783** | 1554 |
| 3 | Lengths | 1.13E-122 | 8089.437 | **7527.859** | 502 |
| 4 | Lengths | 7.61E-82 | 6617.101 | **6243.537** | 412 |
| 1 | Diameter | 1.01E-16 | 5585.737 | **5512.068** | 1016 |
| 2 | Diameter | 0 | 9607.744 | **7700.496** | 1554 |
| 3 | Diameter | 8.76E-28 | **2458.498** | 2583.102 | 502 |
| 4 | Diameter | 2.35E-34 | **1998.699** | 2153.566 | 412 |

above a certain size threshold, were consistent with the elastic similarity model [16]. However, exponents fit to the entire dataset are clearly more consistent with *CFS*. This likely reflects the numerical abundance of the smaller branches and their strong influence on regression fits.

The elastic similarity model (independent of its use within the WBE framework), has been evaluated by a number of authors and most empirical relationships show non-linearity in the allometry of height to DBH [2,21,26,27]. This suggests that there may be some scale (trunk/branch size) at which maintaining elastic similarity becomes important for trees. Recent work examining height to diameter relationships within and across North American tree species has shown that there is indeed a trend toward exponent values closer to those predicted by elastic similarity, particularly among very tall species with strong apical dominance [17]. However, among species with small mature stature, the range of observed exponents is large with the central tendency approaching the *CFS* expectations as trees get smaller.

There are many differences between the *FFS* and WBE models [12]. Among the more important differences is with respect to the scaling of branch lengths. The WBE model assumes that tree branching structure is self-similar (fractal) throughout. In empirical data this would be supported by radius and length ratios clustered around numerical values ($r_{k+1}/r_k = 0.7071$ and $l_{k+1}/l_k = 0.794$), and frequency distributions of lengths and radii exhibiting power-law behavior [4]. In contrast, the *FFS* model assumes that bifurcation in tree branches occurs in response to the availability of heterogeneously distributed light, which when combined with the process of branch shedding and examined at the level of entire trees, begins to look like a stochastic (Poisson) process, leading to an exponential distribution of branch lengths [28]. Previous analyses of distribution of branch lengths and radii in tree branches [12] and leaves [28] support this modelling approach. We find the exponential distribution of lengths is a better fit than a power law in all four of the trees we examined. This supports the idea that branching can be approximated as a Poisson process with a characteristic scale, which is simply the mean branch length. Mean branch length is expected to differ between species based on their habitat specific growth requirements and adaptations.

Only two of the four distributions of branch radii we examined were better fit by a power law (the other two exponential). Whether this represents an anomaly, or important biological signal is unknown. As branch dimensions below a size threshold (100 mm in length or 1.5 mm in diameter) were not collected in this dataset, subsequent investigations including the entirety of branching networks can inform this question. Data on the frequency distributions of branch lengths and radii are not common, so this remains an important target of inquiry for future investigators.

Previous reviews have identified issues with the WBE model's mathematical underpinnings and the disconnect between model assumptions and empirical data [6,7]. Our work further supports the notion that the WBE model is not a very accurate abstraction of tree branching dimensions and architecture. While the WBE model(s) have been very

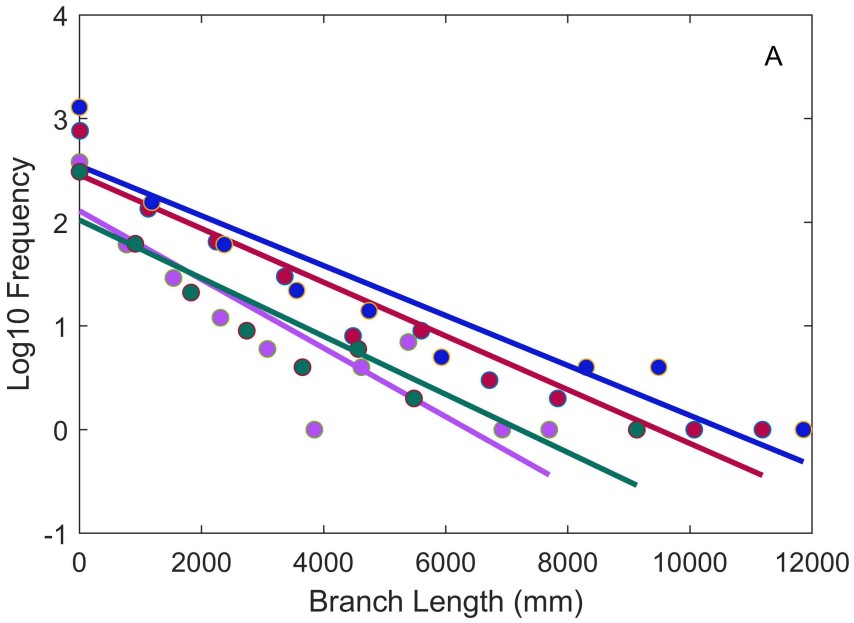

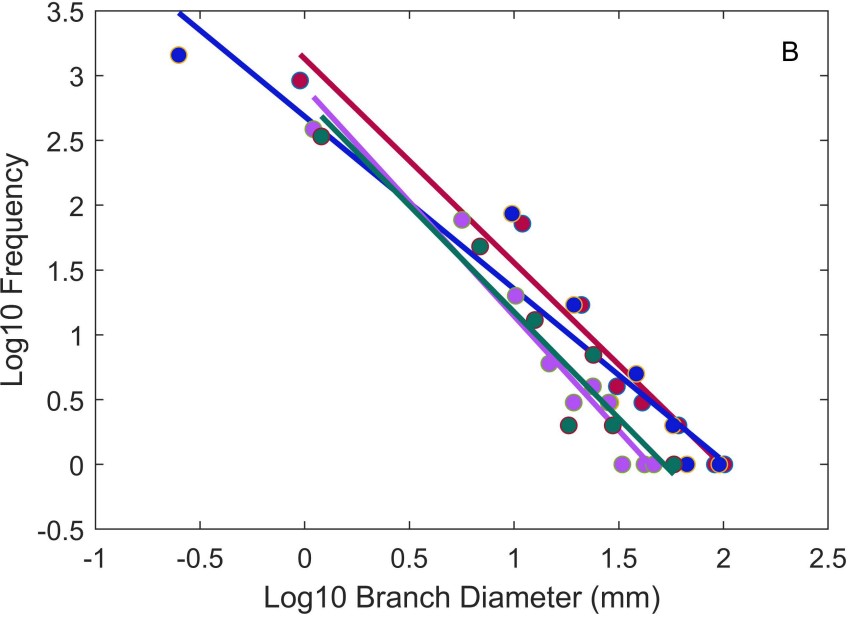

**Fig 3. Frequency distributions for branch lengths (Panel A) and diameters (Panel B) for the four trees we analyzed.** Symbol and line colors correspond to data from each of the four trees. Note the data here are binned for heuristic purposes only, all statistics performed on these frequency distributions utilized bin free, maximum likelihood approaches (see Methods). Lines are SMA fits to binned data and again are for heuristic purposes. Note the length distributions are approximately linear on a log-linear plot, consistent with the expectations for an exponential distribution, while the diameter distributions are approximately linear on log-log plots, consistent with the expectations for a power law.

valuable in catalyzing interest in scaling patterns within and across species, newer models such as *FFS* are likely needed to better characterize the most salient features of plant form underpinning their inter and intraspecific allometric patterns.

## Author contributions

**Conceptualization:** Charles Price.

**Data curation:** Gregory A. Dahle, Jason C. Grabosky.

**Formal analysis:** Charles Price, Gregory A. Dahle, Jason C. Grabosky.

**Methodology:** Charles Price.

**Visualization:** Charles Price.

**Writing – original draft:** Charles Price, Gregory A. Dahle, Jason C. Grabosky.

**Writing – review & editing:** Charles Price, Gregory A. Dahle, Jason C. Grabosky.

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
