## [Decision Letter · Decision Letter 0]

2 May 2025

PONE-D-25-10084Flow similarity model predicts allometric relationships among Acer platanoides branchesPLOS ONE

Dear Dr. Price,

Thank you for submitting your manuscript to PLOS ONE. After careful consideration, we feel that it has merit but does not fully meet PLOS ONE’s publication criteria as it currently stands. Therefore, we invite you to submit a revised version of the manuscript that addresses the points raised during the review process.

**ACADEMIC EDITOR:**Both reviewers found merit in the manuscript. The data was analyzed properly and support the conclusions. Both reviewers made suggestions to improve overall presentation of the manuscript. I would also ask that the authors make their data publicly available upon resubmission of the manuscript, as requested by PLOS ONE. If the authors cannot make it publicly available, please specify the reason. For example:"Data cannot be shared publicly because of [XXX]. Data are available from the XXX Institutional Data Access / Ethics Committee (contact via XXX) for researchers who meet the criteria for access to confidential data." ==============================

We look forward to receiving your revised manuscript.

Kind regards,

Renato Filogonio

Academic Editor

PLOS ONE

[National Institute of Food and Agriculture - McIntire Stennis Project # WVA00108]

 [The author(s) received no specific funding for this work.]

3. In the online submission form, you indicated that [Data are available on request from G.A. Dahle].

4. We noted in your submission details that a portion of your manuscript may have been presented or published elsewhere. [The data we used in our analysis were originally analyzed in: Dahle G, A, Grabosky J, C. Allometric patterns in Acer platanoides (Aceraceae) branches. Trees. 2010;(24):321–6. However, our use of the data and the patterns we find are entirely novel.] Please clarify whether this conference proceeding or publication was peer-reviewed and formally published. If this work was previously peer-reviewed and published, in the cover letter please provide the reason that this work does not constitute dual publication and should be included in the current manuscript.

Reviewers' comments:

Reviewer's Responses to Questions

**Comments to the Author**

1. Is the manuscript technically sound, and do the data support the conclusions?

Reviewer #1: Yes

Reviewer #2: Yes

2. Has the statistical analysis been performed appropriately and rigorously? 

Reviewer #1: Yes

Reviewer #2: Yes

3. Have the authors made all data underlying the findings in their manuscript fully available?

Reviewer #1: Yes

Reviewer #2: No

4. Is the manuscript presented in an intelligible fashion and written in standard English?

Reviewer #1: Yes

Reviewer #2: Yes

5. Review Comments to the Author

Reviewer #1: General comments:

This study tests multiple models regarding the scaling of plant growth form, an approach that I enthusiastically endorse. Good job! I only have relatively minor comments given below.

Specific comments:

L 43-44: Another commonly used physical approach is based on thermodynamics (for recent studies, see e.g., Attorre et al. 2019; Popovic and Minceva 2021; Sabater and Skene 2022; etc.).

L 86-87: However, it apparently does not apply to tracheal branching systems in insects (Aitkenhead et al. 2020), so Da Vinci’s rule is not universally applicable.

L 96-97: Perhaps reword “The 6th column the expected curvature when plotting the relationship in the third column” as “The 6th column refers to the expected curvature when plotting the relationship in the third column”.

L 133: Perhaps reword “The diameter distribution better fit by a power law model.” as “The diameter distribution will be better fit by a power law model.”.

L 134-138: This useful approach follows the method of multiple working hypotheses (Chamberlin 1965), which has been advocated for studies of biological scaling (Glazier 2014).

L 150-151: SMA is preferred when the error of the Y and X variables are approximately equal, which is probably true for the present study. If the error is greater for the Y versus X variable, least-squares regression (LSR) analysis is preferred (see Smith 2009).

L 173-174: Perhaps reword “by fitting both exponential and power law to each data array” to “by fitting both exponential and power functions to each data array.”

L 217-223: Please explain color code.

L 235: Change “in” to “by”,

L 247: Insert “the” before “medium”.

Literature cited:

Aitkenhead, I. J., Duffy, G. A., Devendran, C., Kearney, M. R., Neild, A., & Chown, S. L. (2020). Tracheal branching in ants is area-decreasing, violating a central assumption of network transport models. PLoS Computational Biology, 16(4), e1007853.

Attorre, F., Sciubba, E., & Vitale, M. (2019). A thermodynamic model for plant growth, validated with Pinus sylvestris data. Ecological Modelling, 391, 53-62.

Chamberlin, T.C. (1965). The method of multiple working hypotheses. Science, 148, 754–759.

Glazier, D. S. (2014). Metabolic scaling in complex living systems. Systems, 2(4), 451-540.

Popovic, M., & Minceva, M. (2021). Standard thermodynamic properties, biosynthesis rates, and the driving force of growth of five agricultural plants. Frontiers in Plant Science, 12, 671868.

Sabater, B., & Skene, K. R. (2022). Plant Thermodynamics. Frontiers in Plant Science, 13, 932245.

Smith, R. J. (2009). Use and misuse of the reduced major axis for line‐fitting. American Journal of Physical Anthropology: The Official Publication of the American Association of Physical Anthropologists, 140(3), 476-486.

Reviewer #2: This is a short and well-written paper, easy to follow and pleasant to read.

In this ms, the authors evaluated the predictions of the flow similarity model and WBE model on branching architecture using 3,484 individual stem internodes across four Acer platanoides trees. They evaluated that the analysis of allometric exponents, direction of curvature, area-preserving assumption by area ratios, and frequency distributions of branch lengths/diameters are almost consistent with the model’s predictions.

The introduction is presented well, with the research questions using recent scientific articles. Methodology is presented in concise form. Some clarifications are needed. The results are briefly presented and discussed well by posing future investigations. Overall, the structure and flow of the manuscript are good.

See comments,

Major:

1. The introduction and discussion should be enhanced a bit to ensure a clear narrative flow for a multidisciplinary audience.

2. How the surface area (SA) and volume (V) are calculated. Is it through on a cylindrical assumption? This should be briefly included in the methodology.

3. Page 19, line 275. In the discussion, authors showed that the exponential distribution of lengths is a better fit than a power law in all trees examined. However, the authors claim that stochastic branching can be approximated with a characteristic scale, which is simply the mean branch length. How do authors arrive at a scale simply the mean branch length?

Minor:

1. Use author abbreviations for scientific names of plants at least at the first use in the manuscript.

2. The usage of elastic similarity and the WBE model in the manuscript often confuses the general readers.

Authors should clarify that the elastic similarity model is related to the West-Brown-Enquist (WBE) model (through a structural and hydraulic relation), and thus considered as same in the manuscript. Mention the elastic similarity (WBE) at its first usage.

3. Location and time are fundamental information for methodology. Please include the time, in years, when the data was collected.

4. Page 9, line 82-85. The sentence is confusing. Authors have to describe either, in which the total cross-sectional area of daughter branches (subscript D) equals that of the parent branch (subscript P) on either side of a branch bifurcation, or in which their ratio equals one.

5. Page 17. Line 247. The DBH’s range should be included in the Materials and Methods.

Typos:

1. Page 14, line 170. Capitalize the subscripts as daughter (D) and parent (P).

2. Page 14, line 173. Change radii to diameters (though both mean the same), as the authors mentioned the term diameter in introduction prediction 5 (line 133).

3. Page 15, line 196. Change accordingly. Not area rations. Ratios.

4. The authors describe the AIC scores of the length and diameters for the exponential and power-law models in the text. However, they mentioned it in Table 3 as Radii. Although they mean the same thing, they can be changed for the reader's clarity.

5. Page 17. Line 243. The variability increases with decreasing branch size. Change it to branch radius or diameter. Use of branch size may be confused with length, radius, or any other attribute.

6. PLOS authors have the option to publish the peer review history of their article (what does this mean?). If published, this will include your full peer review and any attached files.

Reviewer #1: **Yes: **Douglas S. Glazier

Reviewer #2: No

---

## [Author Response · Author response to Decision Letter 1]

19 Jun 2025

PONE-D-25-10084

Flow similarity model predicts allometric relationships among Acer platanoides branches

PLOS ONE

Dear Dr. Price,

Thank you for submitting your manuscript to PLOS ONE. After careful consideration, we feel that it has merit but does not fully meet PLOS ONE’s publication criteria as it currently stands. Therefore, we invite you to submit a revised version of the manuscript that addresses the points raised during the review process.

ACADEMIC EDITOR:

Both reviewers found merit in the manuscript. The data was analyzed properly and support the conclusions. Both reviewers made suggestions to improve overall presentation of the manuscript. I would also ask that the authors make their data publicly available upon resubmission of the manuscript, as requested by PLOS ONE. If the authors cannot make it publicly available, please specify the reason. For example:

"Data cannot be shared publicly because of [XXX]. Data are available from the XXX Institutional Data Access / Ethics Committee (contact via XXX) for researchers who meet the criteria for access to confidential data."

Thanks for this. The data have been posted to an online repository: https://doi.org/10.7282/00000503

We have included reference to this on line 154.

We look forward to receiving your revised manuscript.

Kind regards,

Renato Filogonio

Academic Editor

PLOS ONE

Thanks very much for your time and attention on our manuscript. We have incorporated the recommended changes and include a tracked changes version in our revision. We respond to each of the reviewer’s comments below.

Line references refer to the clean version submitted without tracked changes.

[National Institute of Food and Agriculture - McIntire Stennis Project # WVA00108]

[The author(s) received no specific funding for this work.]

3. In the online submission form, you indicated that [Data are available on request from G.A. Dahle].

4. We noted in your submission details that a portion of your manuscript may have been presented or published elsewhere. [The data we used in our analysis were originally analyzed in: Dahle G, A, Grabosky J, C. Allometric patterns in Acer platanoides (Aceraceae) branches. Trees. 2010;(24):321–6. However, our use of the data and the patterns we find are entirely novel.] Please clarify whether this conference proceeding or publication was peer-reviewed and formally published. If this work was previously peer-reviewed and published, in the cover letter please provide the reason that this work does not constitute dual publication and should be included in the current manuscript.

The data were originally analyzed in Dahle and Grabosky (2010) which was peer reviewed. However, none of the patterns we consider in our current manuscript were examined in this previous work, thus this does not constitute dual publication.

Reviewers' comments:

Reviewer's Responses to Questions

Comments to the Author

1. Is the manuscript technically sound, and do the data support the conclusions?

Reviewer #1: Yes

Reviewer #2: Yes

2. Has the statistical analysis been performed appropriately and rigorously?

Reviewer #1: Yes

Reviewer #2: Yes

3. Have the authors made all data underlying the findings in their manuscript fully available?

Reviewer #1: Yes

Reviewer #2: No

The data have been posted to an online repository: https://doi.org/10.7282/00000503

4. Is the manuscript presented in an intelligible fashion and written in standard English?

Reviewer #1: Yes

Reviewer #2: Yes

5. Review Comments to the Author

Reviewer #1: General comments:

This study tests multiple models regarding the scaling of plant growth form, an approach that I enthusiastically endorse. Good job! I only have relatively minor comments given below.

Thanks Douglas for your time and attention reviewing our article! We advocate testing multiple models as well. My (Price) first attempt at this was Price et al. 2009 Ecology Letters (cited in manuscript), and I’ve tried to incorporate this approach whenever possible.

Specific comments:

L 43-44: Another commonly used physical approach is based on thermodynamics (for recent studies, see e.g., Attorre et al. 2019; Popovic and Minceva 2021; Sabater and Skene 2022; etc.).

We’ve added the following sentence on line 77:

“Additional models based on thermodynamics have been published that also warrant consideration [9, 10].”

We’ve added references for Attore and Popovic and Minceva. The Sabater and Skene article is an editorial so we left it out.

L 86-87: However, it apparently does not apply to tracheal branching systems in insects (Aitkenhead et al. 2020), so Da Vinci’s rule is not universally applicable.

No argument here. The physiological differences between Insect tracheal systems and plant hydraulics are so great that we wouldn’t even think to make such a link. In contrast to WBE, the flow similarity model is expected to apply strictly to plants.

L 96-97: Perhaps reword “The 6th column the expected curvature when plotting the relationship in the third column” as “The 6th column refers to the expected curvature when plotting the relationship in the third column”.

Good suggestion, we’ve renamed the column “Expected Curvature”.

L 133: Perhaps reword “The diameter distribution better fit by a power law model.” as “The diameter distribution will be better fit by a power law model.”.

Done (line 138)!

L 134-138: This useful approach follows the method of multiple working hypotheses (Chamberlin 1965), which has been advocated for studies of biological scaling (Glazier 2014).

We’ve included the following sentence at the end of the paragraph (line 143):

“This approach follows the methods of multiple working hypotheses [20] which has been advocated for studies of biological scaling [21, 22].”

L 150-151: SMA is preferred when the error of the Y and X variables are approximately equal, which is probably true for the present study. If the error is greater for the Y versus X variable, least-squares regression (LSR) analysis is preferred (see Smith 2009).

We’re familiar with the paper you reference. Smith says (2nd paragraph of first page), “In contrast OLS is indicated when the values for the X-axis are known without error (i.e., all the error can be attributed to the Y-axis measurements).” Its actually hard to imagine when this will occur in data of this type.

L 173-174: Perhaps reword “by fitting both exponential and power law to each data array” to “by fitting both exponential and power functions to each data array.”

Good catch, thanks!

L 217-223: Please explain color code.

Good suggestion, we’ve added the following sentence to the caption (line 230): “Symbol and line colors correspond to data from each of the four trees.”

L 235: Change “in” to “by”,

Done.

L 247: Insert “the” before “medium”.

Good catch, done.

Literature cited:

Aitkenhead, I. J., Duffy, G. A., Devendran, C., Kearney, M. R., Neild, A., & Chown, S. L. (2020). Tracheal branching in ants is area-decreasing, violating a central assumption of network transport models. PLoS Computational Biology, 16(4), e1007853.

Attorre, F., Sciubba, E., & Vitale, M. (2019). A thermodynamic model for plant growth, validated with Pinus sylvestris data. Ecological Modelling, 391, 53-62.

Chamberlin, T.C. (1965). The method of multiple working hypotheses. Science, 148, 754–759.

Glazier, D. S. (2014). Metabolic scaling in complex living systems. Systems, 2(4), 451-540.

Popovic, M., & Minceva, M. (2021). Standard thermodynamic properties, biosynthesis rates, and the driving force of growth of five agricultural plants. Frontiers in Plant Science, 12, 671868.

Sabater, B., & Skene, K. R. (2022). Plant Thermodynamics. Frontiers in Plant Science, 13, 932245.

Smith, R. J. (2009). Use and misuse of the reduced major axis for line‐fitting. American Journal of Physical Anthropology: The Official Publication of the American Association of Physical Anthropologists, 140(3), 476-486.

Reviewer #2: This is a short and well-written paper, easy to follow and pleasant to read.

In this ms, the authors evaluated the predictions of the flow similarity model and WBE model on branching architecture using 3,484 individual stem internodes across four Acer platanoides trees. They evaluated that the analysis of allometric exponents, direction of curvature, area-preserving assumption by area ratios, and frequency distributions of branch lengths/diameters are almost consistent with the model’s predictions.

The introduction is presented well, with the research questions using recent scientific articles. Methodology is presented in concise form. Some clarifications are needed. The results are briefly presented and discussed well by posing future investigations. Overall, the structure and flow of the manuscript are good.

Thanks very much for your time and attention reading our manuscript. Your comments have help bring clarity to our presentation so we very much appreciate it!

See comments,

Major:

1. The introduction and discussion should be enhanced a bit to ensure a clear narrative flow for a multidisciplinary audience.

We have gone through the introduction and discussion with an eye toward making it more accessible to a general audience. It is always a challenge to find the right balance with this, assuming a certain level of scientific literacy in your audience, but not too much. Hopefully we have made it a bit easier to understand.

See lines: 65, 79, 84, 101, 104, 110, 111, 112, 115, 119, 129, 267, 297

2. How the surface area (SA) and volume (V) are calculated. Is it through on a cylindrical assumption? This should be briefly included in the methodology.

Good suggestion. We have added the following sentence on line 154:

“The surface area and volume for each branch was calculated from diameter and length using standard formula based on the assumption that each branch can be approximated as a cylinder.”

3. Page 19, line 275. In the discussion, authors showed that the exponential distribution of lengths is a better fit than a power law in all trees examined. However, the authors claim that stochastic branching can be approximated with a characteristic scale, which is simply the mean branch length. How do authors arrive at a scale simply the mean branch length?

The scale parameter of an exponential distribution is just the mean of the distribution which is equal to the reciprocal of the rate parameter. We covered this in more detail in Price et al. 2011 which we cite in the manuscript. We appreciate you drawing attention to this as it is more accurate to describe the exponential distribution emerging from a Poisson process than a stochastic process (a Poisson process is a type of stochastic process). We’ve changed line 286 accordingly.

Minor:

1. Use author abbreviations for scientific names of plants at least at the first use in the manuscript

---

## [Decision Letter · Decision Letter 1]

24 Jun 2025

PONE-D-25-10084R1Flow similarity model predicts allometric relationships among Acer platanoides branchesPLOS ONE

Dear Dr. Price,

Thank you for submitting your manuscript to PLOS ONE. After careful consideration, we feel that it has merit but does not fully meet PLOS ONE’s publication criteria as it currently stands. Therefore, we invite you to submit a revised version of the manuscript that addresses the points raised during the review process.

**ACADEMIC EDITOR:**

Both reviewers were satisfied with the author's responses but had a few minor suggestions that should be addressed before acceptance of the manuscript. 

We look forward to receiving your revised manuscript.

Kind regards,

Renato Filogonio

Academic Editor

PLOS ONE

Journal Requirements:

Reviewers' comments:

Reviewer's Responses to Questions

**Comments to the Author**

1. If the authors have adequately addressed your comments raised in a previous round of review and you feel that this manuscript is now acceptable for publication, you may indicate that here to bypass the “Comments to the Author” section, enter your conflict of interest statement in the “Confidential to Editor” section, and submit your "Accept" recommendation.

Reviewer #1: (No Response)

Reviewer #2: All comments have been addressed

2. Is the manuscript technically sound, and do the data support the conclusions?

Reviewer #1: Yes

Reviewer #2: Yes

3. Has the statistical analysis been performed appropriately and rigorously? 

Reviewer #1: Yes

Reviewer #2: Yes

4. Have the authors made all data underlying the findings in their manuscript fully available?

Reviewer #1: Yes

Reviewer #2: Yes

5. Is the manuscript presented in an intelligible fashion and written in standard English?

Reviewer #1: Yes

Reviewer #2: Yes

6. Review Comments to the Author

Reviewer #1: I am largely satisfied with the authors’ responses to my comments on a previous version of this manuscript.

However, I believe that the authors have misunderstood the paper by Smith (2009) regarding the use of OLS versus RMA regression. In their response to one of my previous comments they state: “Smith says (2nd paragraph of first page), “In contrast OLS is indicated when the values for the X-axis are known without error (i.e., all the error can be attributed to the Y-axis measurements).””. Note that this is not Smith’s view but a view that Smith says many investigators use to justify using RMA over OLS. Smith says that this should not be the only criterion. He argues for considering whether the scaling relationship is the same regardless of which variable is the Y versus X. If so, use RMA. If not use OLS. In any case, others point out that even when the X variable has measurement error, OLS should be preferred over RMA if the Y variable is measured with three times more error than that of the X variable (e.g., McArdle 1988; White 2011; Kilmer & Rodríguez 2017). This is because RMA assumes that the Y and X errors are equal (Smith 2009). Kilmer & Rodríguez (2017) also argue that OLS should be preferred over RMA when the measurement errors for both Y and X are relatively low. I hope that this comment is clearer and more helpful than my previous comment. Please see papers listed below for more information.

Kilmer, J. T., & Rodríguez, R. L. (2017). Ordinary least squares regression is indicated for studies of allometry. J. Evol. Biol. 30, 4-12.

McArdle, B.H. (1988). The structural relationship—Regression in biology. Can. J. Zool. 66, 2329–2339.

White, C.R. (2011). Allometric estimation of metabolic rates in animals. Comp. Biochem. Physiol. A Mol. Integr. Physiol. 158, 346–357.

Doug Glazier

Reviewer #2: Thank you, authors, for the revision. Authors have adequately addressed both my major and minor concerns regarding the ms. I hope the authors agree that these revisions have improved the ms.

Comment:

Page 19. Line 153. Cite (by author name and date) the raw data (dataset) properly from the university library.

Thank you once again.

7. PLOS authors have the option to publish the peer review history of their article (what does this mean?). If published, this will include your full peer review and any attached files.

Reviewer #1: No

Reviewer #2: No

---

## [Author Response · Author response to Decision Letter 2]

2 Jul 2025

Second response to reviewers’ comments

Thanks again to the handling editor and reviewers for their comments on our manuscript. There were two follow up comments which we address below.

Reviewer #1: I am largely satisfied with the authors’ responses to my comments on a previous version of this manuscript.

However, I believe that the authors have misunderstood the paper by Smith (2009) regarding the use of OLS versus RMA regression. In their response to one of my previous comments they state: “Smith says (2nd paragraph of first page), “In contrast OLS is indicated when the values for the X-axis are known without error (i.e., all the error can be attributed to the Y-axis measurements).””. Note that this is not Smith’s view but a view that Smith says many investigators use to justify using RMA over OLS. Smith says that this should not be the only criterion. He argues for considering whether the scaling relationship is the same regardless of which variable is the Y versus X. If so, use RMA. If not use OLS. In any case, others point out that even when the X variable has measurement error, OLS should be preferred over RMA if the Y variable is measured with three times more error than that of the X variable (e.g., McArdle 1988; White 2011; Kilmer & Rodríguez 2017). This is because RMA assumes that the Y and X errors are equal (Smith 2009). Kilmer & Rodríguez (2017) also argue that OLS should be preferred over RMA when the measurement errors for both Y and X are relatively low. I hope that this comment is clearer and more helpful than my previous comment. Please see papers listed below for more information.

Kilmer, J. T., & Rodríguez, R. L. (2017). Ordinary least squares regression is indicated for studies of allometry. J. Evol. Biol. 30, 4-12.

McArdle, B.H. (1988). The structural relationship—Regression in biology. Can. J. Zool. 66, 2329–2339.

White, C.R. (2011). Allometric estimation of metabolic rates in animals. Comp. Biochem. Physiol. A Mol. Integr. Physiol. 158, 346–357.

Doug Glazier

Thanks for your detailed comments on this Doug. I (Price) have done a pretty deep dive on the issue as well. I agree with everything you’ve said here regarding the choice of SMA vs OLS regression. The quote from Smith did catch my eye, but I see your point that Smith was presenting the prevailing view not his view.

One challenge is that the error is often unknown a priori and we make some assumptions (which are usually probably reasonable) about which variables have more relative error. Repeated measures could start to get at this but its not the standard in the field.

If I understand your comments correctly, you don’t have an issue with the statistics we used. Your original comment was, “SMA is preferred when the error of the Y and X variables are approximately equal, which is probably true for the present study.” Thus, in your view our stats are fine, you’re just making sure we understand Smith’s viewpoint on the issue which we now do.

Thanks for your careful reading of our manuscript!

Reviewer #2: Thank you, authors, for the revision. Authors have adequately addressed both my major and minor concerns regarding the ms. I hope the authors agree that these revisions have improved the ms.

They have indeed! Thanks again!

Comment:

Page 19. Line 153. Cite (by author name and date) the raw data (dataset) properly from the university library.

We’ve changed the reference as requested!

Thank you once again.

Thank you!

Charles A. Price, on behalf of my coauthors

---

## [Editor Report · Decision Letter 2]

4 Jul 2025

Flow similarity model predicts allometric relationships among Acer platanoides L. branches

PONE-D-25-10084R2

Dear Dr. Price,

We’re pleased to inform you that your manuscript has been judged scientifically suitable for publication and will be formally accepted for publication once it meets all outstanding technical requirements.

Kind regards,

Renato Filogonio

Academic Editor

PLOS ONE
---

## [Editor Report · Acceptance letter]

PONE-D-25-10084R2

PLOS ONE

Dear Dr. Price,

I'm pleased to inform you that your manuscript has been deemed suitable for publication in PLOS ONE. Congratulations! Your manuscript is now being handed over to our production team.

Kind regards,

on behalf of

Dr. Renato Filogonio

Academic Editor

PLOS ONE